# The views, perspectives, and experiences of academic researchers with data sharing and reuse: A meta-synthesis

Laure Perrier[1]*, Erik Blondal[2], Heather MacDonald[3]

**1** University of Toronto Libraries, University of Toronto, Toronto, Ontario, Canada, **2** Institute of Health Policy, Management and Evaluation, University of Toronto, Toronto, Ontario, Canada, **3** MacOdrum Library, Carleton University, Ottawa, Ontario, Canada

☯ These authors contributed equally to this work.
\* l.perrier@utoronto.ca

**Data Availability Statement:** The data are available from the Zenodo Repository, DOI: doi.org/10.5281/zenodo.3258850.

**Funding:** The authors received no specific funding for this work.

## Abstract

### Background

Funding agencies and research journals are increasingly demanding that researchers share their data in public repositories. Despite these requirements, researchers still withhold data, refuse to share, and deposit data that lacks annotation. We conducted a meta-synthesis to examine the views, perspectives, and experiences of academic researchers on data sharing and reuse of research data.

### Methods

We searched the published and unpublished literature for studies on data sharing by researchers in academic institutions. Two independent reviewers screened citations and abstracts, then full-text articles. Data abstraction was performed independently by two investigators. The abstracted data was read and reread in order to generate codes. Key concepts were identified and thematic analysis was used for data synthesis.

### Results

We reviewed 2005 records and included 45 studies along with 3 companion reports. The studies were published between 2003 and 2018 and most were conducted in North America (60%) or Europe (17%). The four major themes that emerged were data integrity, responsible conduct of research, feasibility of sharing data, and value of sharing data. Researchers lack time, resources, and skills to effectively share their data in public repositories. Data quality is affected by this, along with subjective decisions around what is considered to be worth sharing. Deficits in infrastructure also impede the availability of research data. Incentives for sharing data are lacking.

### Conclusion

Researchers lack skills to share data in a manner that is efficient and effective. Improved infrastructure support would allow them to make data available quickly and seamlessly. The

**Competing interests:** The authors have declared that no competing interests exist.

lack of incentives for sharing research data with regards to academic appointment, promotion, recognition, and rewards need to be addressed.

## Introduction

Research communities, including funding agencies and scholarly journals, have moved towards greater access to data through the development of policies that promote data sharing [1–4]. Examples include the development of data sharing requirements for clinical trials by the International Committee of Medical Journal Editors [5], the creation of a data repository for all researchers working towards a solution to the Zika virus so that all data is published as soon as it becomes available [6], and large funding bodies such as the Bill & Melinda Gates Foundation implementing strong open data policies [7].

These global developments require more researchers to share their data and make it available for reuse. Proponents for open data maintain that it offers the opportunity for others to freely reuse data, makes research more reproducible, uses public funds more effectively, and expands the potential to combine data sets for increased statistical power or creating new knowledge [8]. Sharing data is routine and embedded into the research process for some disciplines such as genomics and astronomy [9–10]. However, in many fields data produced by researchers has traditionally only been shared at the discretion of the principal investigator upon request, and otherwise kept in filing cabinets or on hard drives. These global shifts around research data have left some feeling uneasy and argue that those who generate the data own the data, certain studies (e.g., those with human subjects) require protection that may be difficult to assure with open data, and data sharing puts an increased administrative burden upon researchers [11]. There are also concerns of the inequity of a career built on data reuse versus the hard work of writing grants, being 'scooped', or being falsely discredited [11–12].

Although funding agencies, institutions, and journals have implemented policies on data sharing and archiving, these practices have not produced the anticipated results. Researchers still withhold data [13], refuse to share data upon request [14–15], publish without data availability statements [16], and fail to put their data into repositories [16] even after agreeing to share their data when publishing a journal article. Problems encountered when data is retrieved from repositories include inadequate annotation [17], limited structured data (Marwick), and incomplete specifications for data processing and analysis [17]. To gain insights on these behaviors, it is important to understand researchers' perspectives. In this study, we report on researchers' views and experiences on data sharing and reuse.

### Aim

Our metasynthesis focuses on the individuals conducting research, and synthesizes the available qualitative literature that examines academic researchers and data sharing. This study addresses the question: what are the views, perspectives, and experiences of academic researchers on data sharing and reuse of research data?

## Materials and methods

A protocol was developed and is available upon request to the authors. Although the PRISMA statement has not been modified for meta-syntheses, it was used to guide the reporting of this review and can be viewed in S1 Appendix.

## Types of studies

This is a metasynthesis of qualitative primary studies. Qualitative research seeks to discover how people perceive and experience the world around them [18]. Direct communication (e.g., interviews, focus groups) or observation are used to explore people's perceptions. Data is explored using qualitiatve analytical methods and findings are then presented narratively using thick description rather than through numbers [19]. Thick description presents the findings as they were interpreted or explained by the authors as opposed to simply providing descriptive summaries of each study [20]. This provides the opportunity to translate the findings into a richer, more complete understanding of a phenomenon [21]. We included studies that reported qualitative methodologies and utilized qualitative methods for data analysis. Studies that collected data using qualitative methods but did not use qualitative analysis (e.g., surveys with open-ended questions that used descriptive statistics) were excluded. Mixed methods studies were included if it was possible to retrieve findings exclusively from the qualitative research.

**Identification of studies.**   The studies used for our meta-synthesis were derived from two sources. The first source was a dataset [22] generated from a scoping review on research data management in academic institutions [23] which provided records from inception to April 2016. The purpose of the scoping review was to describe the volume, topics, and methodological nature of the existing research literature on research data management as it specifically related to academic institutions. The search strategy included the terms data sharing, sharing research, data reuse, and research reuse, along with spelling variations and wildcards to ensure all relevant records were captured. The second source for data came from re-running the literature searches from the scoping review with the addition of a validated qualitative search filter [24] in order to retrieve current records from April 2016 to October 2018. When the searches were conducted for the update, four of the original literature databases were unavailable and were replaced with comparable platforms upon consultation with subject matter specialists. Both the original and the updated search included a total of 40 literature databases representing a wide range of disciplines. The search strategy for MEDLINE can be found in S2 Appendix and the full search strategies for the other databases can be obtained by contacting the author. S3 Appendix lists all the literature databases searched. As well, the grey literature, conference proceedings, and a search of the reference list of all included studies were completed. No restrictions were placed on publication date or language. Thus, a comprehensive search of the literature was conducted.

**Eligibility criteria.**   We aimed to identify all studies that investigated the views, perspectives, and experiences of academic researchers with data sharing and reuse. Studies were included if they were original research and reported qualitative methodologies, specifically focus groups or interviews. Studies had to include researchers (full- or part-time) conducting studies in academic institutions. We defined an academic institution as a higher education degree-granting organization dedicated to education and research. If studies included a mixed population, 50% or more of the total sample had to be researchers from academic institutions in order to be eligible for inclusion. Data sharing is defined as the practice of making data available for reuse [25]; reuse is defined as the use of content outside of its original intention [25]. Examples of this include depositing data into a digital repository or publishing raw data. Mixed methods studies that used both qualitative and quantitative methods within the same study were eligible if the qualitative portion met our inclusion criteria.

**Study selection.**   Two investigators independently screened all records from the scoping review dataset in order to identify qualitative studies that met the eligibility criteria. The records from the updated search were assessed for eligibility by two investigators independently

at level 1 (title and abstract) and level 2 (full-text) screening. Discrepancies were resolved by discussion or by a third investigator at every phase of study selection.

**Quality appraisal.** The CASP (Critical Appraisal Skills Programme) Qualitative Checklist [26] was used for quality appraisal of all included studies. The CASP Qualitative Checklist is a 10-item checklist that examines three domains: validity of results, reporting of the results, and value of the research (S4 Appendix). Each study was assessed independently by two investigators and discrepancies were resolved by discussion.

**Data abstraction and analysis.** The authors read and reread all articles to become familiar with each study [27–28]. A data abstraction sheet was developed that included the study characteristics (e.g., year study conducted), participant characteristics (e.g., sample size), and key concepts. Key concepts or interpretations included all findings along with associated quotes from study participants [29]. Thematic analysis using constant comparison was used for data synthesis [30]. An initial set of 10 studies were coded independently by two investigators. These codes were then compared and refined in order to create an initial set of codes that were used going forward. Two investigators continued to independently code the remainder of the studies in duplicate and the team met regularly to discuss and iteratively refine the codes. All discrepancies were resolved through discussion or by a third member of the team. Finally, analytical themes were generated that offered a higher level interpretation beyond a descriptive synthesis [27,31]. The codes were grouped and categorized by making comparisons across articles in order to ensure that we appropriately captured similar themes from multiple studies. Meetings were used to review all constructs and resolve discrepancies, resulting in a refinement of the analytical themes.

## Results

Forty-five studies and three companion reports were included in the review (Fig 1). Included studies are listed in S5 Appendix.

### Study characteristics

The studies included in the review were published during a 15 year period between 2003 and 2018. The most studies were published in 2014 (11 out of 45) and the method for data collection were interviews (37), a combination of interviews/focus groups (5), and focus groups (4). Over half of the studies (27 out of 45) were conducted in the United States. Table 1 provides a summary of study characteristics.

### Findings

We identified four major themes and several sub-themes. The four major themes were data integrity, responsible conduct of research, feasibility of sharing data, and value of sharing data. Themes and sub-themes along with illustrative quotes are summarized in Table 2. Each theme is described below and details are provided for each sub-theme.

**Data integrity.** The theme *data integrity* addresses researcher's perspectives on data that are available from repositories and their expectations around the prospects of reusing this data based on the quality, documentation available, and what individual researchers deemed as worthy of sharing.

Data quality: Researchers acknowledged that although there may be interest or willingness to consider using open data, there would always be people that would not trust the quality of a dataset unless they collected it themselves. They recognized the need to manage expectations with some suggesting that lowering their standards related to data quality may help

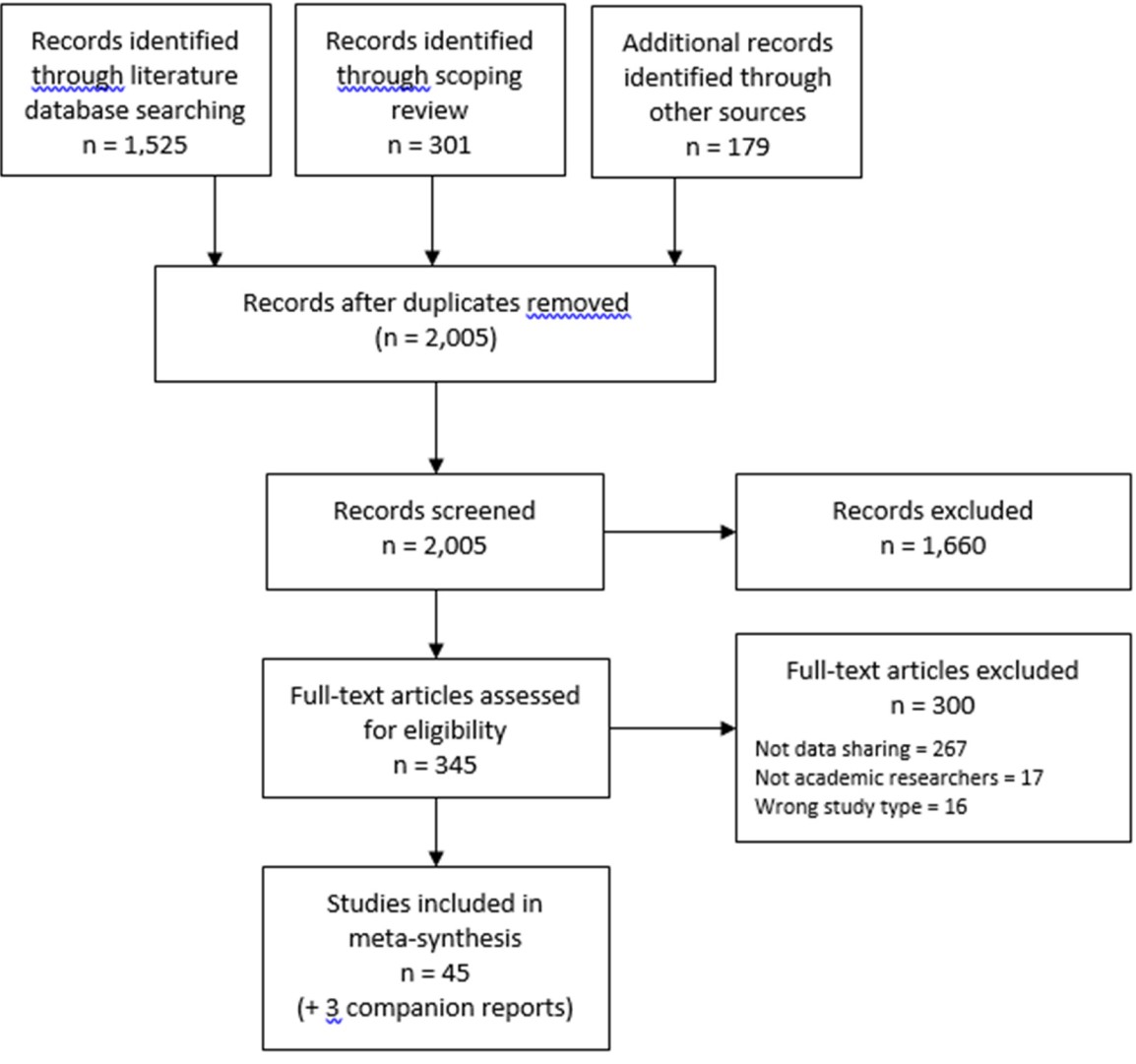

**Fig 1. Study flow diagram.**

with increasing the likelihood of being able to use another researcher's dataset. Nuances around the conditions, context, or materials that are not normally recorded were identified as one of the challenges to data quality. One example was offered in the field of engineering where equipment may not be anchored solidly and produce a variation that would impact the quality of data outputs [42]. The range of skill levels of researchers (e.g., junior versus senior researchers) was flagged as potentially affecting the quality of data collected and it was noted that there was no way of knowing this when reusing a dataset. Similarly, the quality of datasets may also vary depending on the person's intentions or purpose when collecting data. It was felt that data collected with the intention of reporting only to people within their own discipline may look different from data collected for external groups, (i.e., these datasets may include more details).

**Data documentation:** For a dataset to be truly reusable, researchers indicated adequate documentation was necessary including a significant amount of detail and metadata. The importance of contextual information was noted as providing layers of information that offered

**Table 1. Characteristics of included studies.**

| Study | Country | Type (no.) of participants | Data collection | Methodology | Principal experiences explored |
|---|---|---|---|---|---|
| Allard 2012 [32] | Turkey | • Environmental scientist (10) • Information science academic (2) | Interviews | Grounded theory; Analytic induction | Understanding knowledge and attitudes of information science and environmental towards scientific data and information |
| Bamkin 2014 [33] | United Kingdom | Participant (9) | Focus Groups | NR | Identify the opinions of potential users of a policy databank service |
| Broom 2009 [34] | Australia | • Education (12) • Sociology (5) Anthropology (5) • Social Work (5) • Public Health (5) • Psychology (2) Journalism (1) • Politics (2) | Focus Groups | Interpretive qualitative approach | Explore the perceived challenges posed by contemporary innovations in data management, access, and analysis through electronic archiving |
| Carlson 2013 [35] | USA | • PhD student (5) • Masters student (1) | Interviews | NR | Understand grad students practices with data, the challenges they face, and their attitudes toward managing and sharing data |
| Cheah 2015 [36] | Thailand | • Interview participant (15) • Focus group participant (10) | • Interviews • Focus Groups | NR | Understand attitudes and experiences of relevant stakeholders about what constitutes good data sharing practice |
| Colledge 2014 [37] | Switzerland | Stakeholders (includes clinicians, pathologists, lawyers, ethicists, and biobank managers, researchers) (36) | Interviews | Classical qualitative method | Opinions regarding getting consent for sharing samples with biobanks, the role of ethics committees |
| Cragin 2010 [38] | USA | • Agronomy and soil science (5) • Anthropology (3) • Earth and atmospheric science (2) • Geology (3) • Horticulture • and plant science (2) | Interviews | NR | Investigate how data-related scholarly activities vary among disciplines and research communities |
| Delasalle 2013 [39] | United Kingdom | Participant (8) | Interviews | NR | Charts the steps taken and possible ways forward to research data management, providing a typical example of a UK research university's approach in two strands: requirements and support |
| Denny 2015 [40] | South Africa | • Community research support team (2) • Junior research staff (10) • Research manager (4) • Senior researcher (10) • Policy and department manager (3) • Executive member (3) | • Interviews • Focus Groups | Grounded theory; Thematic framework approach | Examine the perceptions, experiences and concerns of research stakeholders about data-sharing practices |
| Diekmann 2012 [41] | USA | Participant (14) | Interviews | NR | Examine data practices of agricultural scientists |
| Faniel 2010 [42] | USA | • Assistant professor (4) • Associate professor (2) • Full professor (6) • PhD student (1) • Postdoctoral student (1) | Interviews | NR | Examine how earthquake engineer researchers assess the reusability of colleagues' experimental data for model validation |
| Faniel 2013 [43] | NR | Participant (22) | Interviews | NR | Examine the needs of archaeological data re-users, particularly the context they need to understand, verify, and trust data others collect during field studies |

*(Continued)*

**Table 1.** (*Continued*)

| Study | Country | Type (no.) of participants | Data collection | Methodology | Principal experiences explored |
|---|---|---|---|---|---|
| Finn 2014 [44] | Multiple countries in Europe, USA | Participant (5) | Interviews | NR | Identify legal and ethical issues relevant to open access to research data, identify examples that illuminate these issues, and identify potential solutions currently being used to address these issues |
| Frank 2015 [45] | USA | Archaeologists (22) Zoologists (27) | Interviews | Not explicitly stated but declares, "combined deductive and inductive approaches" | • Practices and norms affect how archaeologists and zoologists view/understand preservation as it relates to their own research data • External factors influencing attitudes of archaeologists and zoologists toward the feasibility of long-term preservation of research data |
| Hall 2013 [46] | USA | Environmental science (14) | Interviews | Phenomenological approach | Determining where metadata re-use is most common or lacking |
| Henty 2008 [47] | Australia | NR | • Focus Groups • Interviews | NR | Needs related to and provisions of data management infrastructure |
| Higman 2015 [48] | United Kingdom | Participants (librarians, research office staff and IT professionals) (11) | Interviews | Interpretivist perspective | • Relationship between research data management (RDM) and data sharing in formulations of RDM policies • Clarify what is influencing decisions, how different actors are behaving and how networks are being formed |
| Hunt 2018 [49] | USA | Professor (12) Associate professor (8) Assistant professor (4) | Interviews | Grounded theory | To assess the comprehensive information science needs and behaviors of public health research faculty |
| Johnston 2014 [50] | USA | Faculty (1) Graduate student (4) | Interviews | NR | Needs and data management skills required by graduate student in engineering field |
| Johri 2016 [51] | USA | Associate professor (2) Assistant professor (2) Graduate students (2) | Interviews | NR | To get better insights into the current state of data sharing in engineering education and what needs to be done if data sharing is to be supported |
| Kervin 2012 [52] | | PhD student (3) Post-doc (1) Professor (1) | Interviews | NR | How researchers handled data in a research project from start to finish |
| Kim 2012 [53] | USA | Tenured (full and associate) professors (11) Assistant professors (8) Emeritus professor (1) Professor of practice (1) Post-doctoral research Associates (2) Doctoral candidates (2) | Interviews | Inductive approach (mentioned with regards to coding scheme) | Sharing practices in diverse fields and factors motivating or preventing data sharing |
| Lage 2011 [54] | USA | NR | Interviews | Ethnographic | Represent the range of attitudes and needs regarding the type of datasets created, existing data storage and maintenance support, disciplinary culture or personal feelings on data sharing, and receptivity to the library's role in data curation |

(*Continued*)

**Table 1.** (Continued)

| Study | Country | Type (no.) of participants | Data collection | Methodology | Principal experiences explored |
|---|---|---|---|---|---|
| Manion 2009 [55] | USA | Participant (24) | • Interviews • Focus Groups | NR | Collect policy statements, expectations, and requirements from regulatory decision makers at academic cancer centers in the United States; use these statements to examine fundamental assumptions regarding data sharing using data federations and grid computing |
| Marcus 2007 [56] | USA | Interview participant (7) Focus group participant (NR) | Interviews | NR | Capture the practical and conceptual challenges of research in the sciences |
| McGuire 2012 [57] | USA | Investigators (63) Recruits (50) | Interviews | Thematic content analysis | Explore core ethical, legal, and social implication issues that arose during the first phase of the Human Microbiome Project from the perspective of individuals involved in the research |
| McLure 2014 [58] | USA | Researcher (31) | Focus Groups | Thematic analysis | Understanding nature of researcher data sets, their management, need for assistance/ support, library support |
| Murillo 2014 [59] | USA | Participant (14) | Focus Groups | Inductive content analysis | Scientists' perceptions on the topic of data at risk; re-use/sharing; Data At Risk Inventory |
| Noorman 2014 [60] | United Kingdom | Data centre manager, project coordinator, librarian (15) | Interviews | NR | • Focus on challenges faced by institutions, such as archives, libraries, universities, data centers and funding bodies, in making open access to research data possible • Explore current strategies, the remaining barriers and possible solutions for overcoming these barriers |
| Ochs 2017 [61] | USA | NR | Interviews | NR | To examine various aspects of the research life and process of faculty and research staff in the agriculture discipline |
| Oleksik 2012 [62] | United Kingdom | Professor (1) Post-docs (3) PhD students (5) | Interviews | Thematic analysis | Understand the interdependencies of technologies, practices, and artifacts that emerge as part of the scientific activities |
| Pepe 2014 [63] | USA | Astronomers (12) | Interviews | NR | Gather a first-hand account of the needs and challenges of data referencing and archiving in astronomy |
| Read 2015 [64] | USA | Basic scientists (11) Clinical researchers (19) | Interviews | Grounded theory | Obtain information to plan data-related products and services |
| Stamatolos 2016 [65] | NR | Faculty (14) | Interviews | Inductive approach | Seek an understanding of the thinking and practices of a small, but diverse population of faculty researchers regarding data management |
| Stapleton 2017 [66] | USA | Professors (6) Associate professors (4) Assistant professors (4) Non-tenure track research assistant (1) | Interviews | Grounded theory | The research practices of academics in agriculture in order to understand the resources and services these faculty members need to be successful in their teaching and research |
| Sturges 2014 [67] | NR | Participant (23) | Focus Groups | Grounded theory | Views and practices of stakeholders to data sharing |
| Valentino 2015 [68] | USA | Graduate student (5) | Interviews | NR | Allow students to explain their research covering the areas of data analysis, storage, organization, and format, and data back-up practices |

(*Continued*)

**Table 1.** (Continued)

| Study | Country | Type (no.) of participants | Data collection | Methodology | Principal experiences explored |
|---|---|---|---|---|---|
| Van den Eynden 2014 [69] | Europe (Denmark, UK, Germany, Netherlands, Finland) | Participant (22) | Interviews | Comparative analysis | Data sharing practices and motivation for data sharing |
| Van Tuyl 2015 [70] | USA | NR | Interviews | NR | Formalize assessment of research data management practices of researchers at the institution by launching a faculty survey and conducting a number of interviews with researchers |
| Wallis 2013 [71] | USA | Participant (43) | Interviews | NR | Motivation for sharing data; conditions placed on data that is shared; sharing data with others outside own research group; data used that were not generated by a researcher's own group; how are data from external sources is used |
| Williams 2013 [72] | USA | Assistant professor (3) Associate professor (1) Full professor (3) | Interviews | NR | Summarize the participants' reasons for making data publicly available but also describes the challenges that they faced when sharing data |
| Yatcilla 2017 [73] | USA | Agricultural & Biological Engineering (3) Agricultural Economics (1) Agronomy (4) Botany & Plant Pathology (1) Entomology (3) Forestry & Natural Resources (1) Youth Development & Agricultural Education (3) Agricultural Administration (1) | Interviews | NR | To understand the resources and services these faculty members (agriculture) need to be successful in their research and teaching |
| Yoon 2014 [74] | USA | Faculty (17) Research associate (2) | Interviews | Inductive approach | Enhance our understanding of trust in repositories from the users' point of view |
| Yoon 2017 [75] | USA | PhD students Postdocs Professors Research scientists *No numbers provided* | Interviews | Interpretive qualitative approach | To investigate reusers' trust beyond trust formation and tracks those changes to trust that happen during the experiences of using data |
| Zimmerman 2003 [76] | NR | Ecologists (13) Data managers (4) | Interviews | Inductive approach | Experiences of ecologists who use shared data |

necessary insight. Providing this was seen as a time- and energy-intensive endeavor for the researcher collecting the data and making it available for reuse. Comprehensive documentation signaled a reliable dataset to researchers that were looking for datasets to reuse.

**What is worth sharing:** When considering their data, researchers varied on what was worth sharing. It spanned from believing the preservation of datasets to be a top priority, to complete lack of interest. For those that felt their data was not worth sharing, they believed their data would be irrelevant after a period of time and nothing more than a "historical curiosity" if it was offered for reuse [56]. Others thought their data had the potential to be useful but had clear views on what was worth sharing and felt it had to have 'scholarly value'. As an example, researchers described biospecimens they had collected as valuable since it provided the opportunity to look at the development of a disease [64].

**Responsible conduct of research.** *The responsible conduct of research* emerged as a theme that encompassed the professional standards, ethical principles, and tacit norms that

**Table 2. Themes derived and illustrative quotes.**

| Theme and sub-theme | Illustrative quotes | Reference |
|---|---|---|
| **Theme: Data Integrity** | | |
| Data quality | There are definitely different comfort levels for people. Some people will forever be confined to studying their own system because they are unable to accept any degree of, you know, sort of taking other people's word—sort of dealing with data that they didn't actually see collected themselves. | Zimmerman 2003 [76] |
| | What had been reported, what had been presented and discussed were, kinda, the best view of the data. [I]n reality, the data did have some problems that weren't apparent until you got deeply inside and started looking. | Yoon 2017 [74] |
| Data documentation | . . .a lot of the contextual data that you need is not provided. | Faniel 2013 [43] |
| | You can tell from the documentation whether or not a research[er] was thorough and careful. | Yoon 2017 [75] |
| | It's so easy to generate this digital data, but if you're not careful how you name things and how you document stuff and making sense of it later, particularly for someone else, is going to be a real challenge. | Yatcilla 2017 [73] |
| What is worth sharing | Am I worried it won't be there in 20 years? No. Am I worried it won't be there in 100? It doesn't matter. By that point, data become irrelevant except as historical curiosity. | Marcus 2007 [56] |
| | Biospecimens are very valuable because they were collected before the disease, so they're good for looking at developing disease. . .I think it could be used for many years. | Read 2015 [64] |
| **Theme: Responsible Conduct of Research** | | |
| Misuse of data | . . .my main concern is I don't want people to misuse it . . . and if I don't have some relationship of trust then I don't know whether they're going to, you know, just go off and do something and never check with me to see, well, was this a good interpretation. | Cragin 2010 [38] |
| | . . .a whole cadre of people whose only job is pilfering other people's stuff, or parasitically using it. | Hunt 2018 [49] |
| Work culture | I completed an NSF grant in December and. . . you have to have now a section that describes what you are going to do with your data. . .Data availability and where you're going to archive it. . . So you're being forced to deal with it now whereas in the past you're like, 'Well it's in my file cabinet. | Frank 2015 [45] |
| | I think perhaps it's just tradition or it's a thing of the past where people have held their data somewhat closely. . . | Ochs 2017 [61] |
| Protecting one's own work / Intellectual property | We all collect samples together in the field, but when you come back to process the samples, people want the data without any understanding or agreement about ownership. | Marcus 2007 [56] |
| | But it's also the notion of intellectual property, isn't it? . . . How are we going to know if other people are picking it up and using it elsewhere, unless they're being absolutely. . . | Broom 2009 [34] |
| Control of data | If someone were to use the data would be good to know, what did they do with it, some form of communication. . . | Johri 2016 [51] |
| | You would have to describe your intended use of the data. And then the people who originally were the researchers who gathered that data, would all have to agree to consent to each application. And so they still retain the control of the data. | Finn 2014 [44] |
| Privacy/Confidentiality/Ethics | If the systems are such that they can get into our data, we might need to think for the first time about being a little bit more circumspect and think about what qualifications we would want to impose . . . I think there would probably be a lot of regulatory compliance pieces we might want to spell out more than we do now. | Manion 2009 [55] |
| | . . .we can never actually, never guarantee confidentiality of all data, because it could be hacked into and we can't anymore say that your data will be anonymous because that is nonsense too, because we are able to bring in so many different kinds of data, . . . that the potential for people to be re-identified or distinguished in the data are quite high. . . | Finn 2014 [44] |
| **Theme: Feasibility of Sharing Data** | | |
| Infrastructure | I do think that from an institutional level there should be a governing body to provide guidance and to enforce policy, and to make policy for all the systems that will interact and handle activity with other institutions. As far as what functions they would dictate, [they] would be all around the authorization, authentication, and accounting of access to that data. | Manion 2009 [55] |
| | It's very easy to see how having a central, university wide, storage and dissemination system for data would be much more cost effective, and probably better executed, than anything we could do ourselves. | McLure 2014 [58] |
| Time/work required | If there's someone in the institute who can [deposit data], instead of individual researchers, that would save lots of our time and [we could] be more productive. . . | Williams 2013 [72] |
| | To be quite honest, the biggest hurdle when you're dealing with genetic data in like depositing . . . the information and the sequence data onto GenBank is associating that with museum specimens or locality data . . .It's really kind of clunky and it really takes a lot of time to do that. | Frank 2015 [45] |

*(Continued)*

**Table 2.** (Continued)

| Theme and sub-theme | Illustrative quotes | Reference |
|---|---|---|
| Skills | We are not thinking too much about data management. We are thinking more about the approach and methodology. . . | Diekmann 2012 [41] |
| | They are resistant to having to learn how to use new tools that make open data and reproducibility easier. They generally kind of just have their process and they feel like they're tested already in terms of their time and their commitment and they don't really want to add this to the list of things that they have to worry about. | Noorman 2014 [60] |
| **Theme: Value of Sharing Data** | | |
| Promote future discovery | . . .there is no sense in collecting data if it can't be used [by other researchers]. | Lage 2011 [54] |
| | We truly believe that sharing data is the right thing to do, simply because the original data we used for this study was not ours. Our study was only possible because other astronomers made their data publicly available in the first place! | Pepe 2014 [63] |
| Researcher perspective | To incentivize data sharing there should be follow-on grants on data analysis and dissemination grant to bring other researchers on board. If NSF changed their model for a year, there is a lot of data out there. I think there has to be some stipulation about who gets authorship when the data is used but I think funding to bring new people on board is essential. There can also be a solicitation focused on secondary analysis. | Johri 2016 [51] |
| | I think one barrier to data sharing is the merit review process within institutions for tenure and promotion; things such as 'how many people accessed your dataset' are not valued. | Johri 2016 [51] |

researchers described when considering data sharing. The five sub-themes under this theme are the misuse of data, protecting one's own work/intellectual property, privacy/confidentiality/ethics, control of data, and work culture.

**Misuse of data:** Researchers expressed concern about the potential for the inappropriate use of their data. This included what was termed as 'fishing expeditions' which involved dredging data with no particular research question in the hopes of stumbling upon possible relationships that could be presented as convincing results. The potential for data to be misunderstood and thus produce inappropriate or misguided conclusions was also considered a possibility, even if researchers reused datasets with a focused research question. The people reusing data were given names such as 'free riders' [42] and it was believed that misunderstood data could lead to false conclusions and ultimately threaten the original work related to the data.

**Protecting one's own work/intellectual property:** Clarity around who owns data, along with intellectual property rights, were raised as issues when sharing data. Collaborations were mentioned as making it difficult to determine ownership since multiple people and institutions were involved. Licensing data was seen as both a potential solution, as well as a potential barrier (e.g., the cost could mean it would not be accessible to all) for providing access to research data. Since data had publication value, this was considered a major deterrent to sharing data. This obvious connection between publications and data made researchers feel that data needed to be protected. Publications were seen as a key research output with a relationship between this and future funding.

**Privacy/confidentiality/ethics:** Privacy and confidentiality were taken seriously when researchers considered their data, particularly when it came to human subjects. When data was shared between institutions, the precautions undertaken were complex. This included considerations such as de-identification, re-identification risks (along with the potential for this to happen unintentionally), consent (e.g., whether re-consent was necessary), and the challenge of future use if the purpose for future use was not pre-defined. Precautions implemented included each institution independently obtaining ethics approval before data was

exchanged. Views were divergent around whether data should be freely accessible with some favoring restrictions on the re-use of data and others indicating it should be freely accessible.

**Control of data:** It was acknowledged that the relationship between research being publicly funded and making data available for public benefit had merit. However, some felt they would like to know who was using their data and for what purpose. It was considered important to have a relationship with the person who wished to reuse data. As well, access should be controlled and only given to those that could be identified as a research professional who were qualified to do research. The level of control ranged from wanting systems in place that would allow them to monitor data sharing with people who were not known to them personally (although this was acknowledged as labor-intensive), to simply wanting to have a list of who was using their data and for what purpose as a minimum level of communication. The need to protect data until publication was considered a deterrent to sharing data and it was necessary to have control over data until this was completed.

**Work culture:** Work culture highlights the beliefs of how research should be conducted that are influenced by shared attitudes, views, and written/unwritten rules developed over time. These normative values were described as being taught to junior researchers by senior academics. In the past, the cultural norm was to rely on informal processes, such as personal relationships, for sharing data. Usually, these were people who were known and trusted either through direct contact or by reputation. As new requirements were introduced by funding agencies and journals, researchers observed changes in their practice of sharing data over time. There was an acknowledgement that a shift in culture that favored a more open view of data was needed. It was also noted that even if researchers were to understand the benefits of sharing data, this transition would not be immediate and that incentives must be identified for researchers in this process.

**Feasibility of sharing data.** The *feasibility of sharing data* examines the ease with which researchers can make their data available to others, along with the related barriers and facilitators. Infrastructure, time/work required, and skills are the three sub-themes that emerged and are described below.

**Infrastructure:** Researchers described the structures and supports that would help ensure data sharing. Infrastructure support included data storage, file migration, and funding for making data available. These challenges involved both local (e.g., individual research labs) and institution-wide settings. Data handling was a fragmented activity managed by researchers who devised their own independent strategies that generally lacked sustainability. One example was the number of people (i.e., students, staff) that rotated through a research lab where everyone was responsible for their own data [62]. Although the lab manager encouraged best practices (e.g., including sufficient documentation), this did not necessarily translate into being adopted and applied. As a result, it was not guaranteed that datasets could be easily shared or be accessible in the future. Appropriate data storage infrastructure and support were associated with good data management, which in turn laid the groundwork for data sharing. An institution-level policy to support data sharing, along with resources, were identified as important to ensure good quality data being deposited and made available.

**Time/work required:** The effort to prepare data for sharing was seen as time-consuming, expensive, and labor-intensive. Barriers included the lack of time to organize the necessary documentation, challenges with repository interfaces, and the lack of resources. For those

that chose to offer their data upon request, the administrative aspect of filling requests for data was considered an added burden.

**Skills:** For many disciplines, data sharing was a new activity that was typically imposed by funding agencies or journals. As a result, researchers were looking for services or resources that would help with this task. The lack of technical skills and knowledge included how to anonymize data, how to create metadata, and unfamiliarity with depositing data into repositories. It was felt that providing open access to data was complex. Providing adequate support may not be feasible given that each discipline had a variety of data types, different amount of data being generated, disparities in what is considered data, and varying norms in research culture.

**Value of sharing data.** The *value of sharing data* theme describes researchers' views on the importance placed on making data available to others. While the sub-theme *promote future discovery* identifies a benefit to society with sharing data, *researchers' perspectives* focused on the benefits to researchers themselves.

**Promote future discovery:** The importance of making data accessible for possible use in the future was understood as a benefit. Those that described proactively sharing data (before they were required to) also noted the importance of sharing computer code as well. There was recognition that research funded by public money should be open and available. It was felt that taxpayers provided an investment and the public deserved a return on their investment. In some instances, researchers were able to identify examples of data sharing that helped promote scientific progress, such as the development of a new drug or containment of a disease. It was felt that data sharing had the potential to move a field forward by closing knowledge gaps and further opening new avenues of investigation.

**Researchers' perspectives:** Data was identified as a research product that helped achieve a goal such as completing a publication and there was the recognition that amongst researchers that data sharing would provide greater accountability and transparency. For those that were already reusing data, its value was recognized as helpful for writing proposals and training students. The importance of providing incentives for sharing data was emphasized with researchers unable to identify significant benefits. Suggestions included creating grants that focused specifically on the reuse of data generated from earlier grants.

**Quality assessment.** The CASP tool, used to assess the quality of studies, identified 27 (out of 45) studies that had seven out of the ten items present. Most studies adequately addressed the methods (41 out of 45 studies) and aims (40 out of 45 studies) (Fig 2). Author reflexivity, which asked if the relationship between the researcher and participants was adequately considered, was not apparent in any of the studies. No studies were excluded due to a low score as this may have eliminated those with relevant and insightful results [77–78].

## Discussion

We conducted a comprehensive review that included 45 studies along with 3 companion reports on the views, perspectives, and experiences of academic researchers on sharing their research data. The National Institutes of Health (NIH) in the United States were one of the first funding agencies to introduce a policy on sharing research data in 2001 [4]. This aligns with beginning to see research published on this topic starting in 2003, along with over half of the studies being conducted in the United States. Our results show that some of the themes and sub-themes offer positive support for sharing data however, most highlight areas of

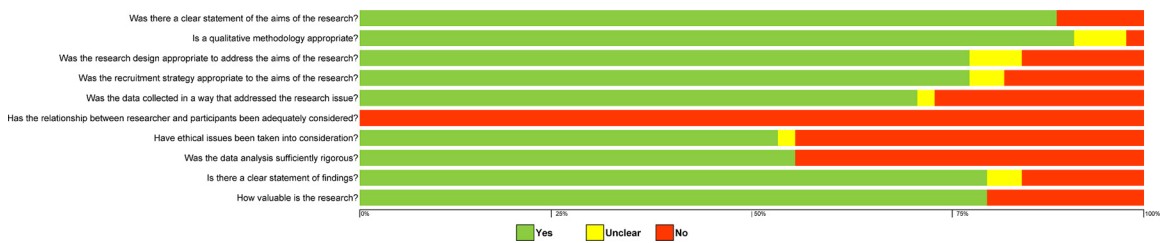

**Fig 2. Quality appraisal of included studies.**

discomfort for researchers. In particular, researchers identified concern with issues related to data quality, misuse of data, protecting data, lack of time and skills, and deficiencies in infrastructure and support.

By default, researchers believed the quality of datasets available for reuse were poor and there is support for this in the literature. Studies assessing data available in public repositories have found incomplete datasets, saved in a way that compromised reuse [79–80]. Researchers who felt their data had value describe using a tacit set of criteria to determine if it had 'scholarly value' [38]. These criteria are based on discretion and would vary from person to person thus adding further to factors that affect the quality of datasets. For researchers who felt their data was not worth sharing, this may be reflected in how they prepare their data for depositing into repositories (e.g., providing poor documentation) and ultimately its final quality.

The lack of supportive infrastructure, lack of time, and skills deficit had an influence on data quality as well as data availability. Researchers indicated that the lack of time and skills impacted the production of sufficient data documentation, creation of suitable metadata, and appropriately anonymized data. They also lacked skills in navigating repository interfaces in order to deposit data. While training and education may address these issues [81], a more effective pathway is to focus energy and resources on creating user-friendly interfaces that allow users to accomplish their goal of depositing datasets as quickly and easily as possible [82]. At an institutional level, the lack of procedures, policies, and guidance contributed to challenges in sharing data. This was particularly true for sensitive data that requires more vetting and scrutiny before sharing. Solutions for this include using a trusted party regulated by an ethics board that manages requests and maintains the de-identified records and original identifiers [55].

Our results show that a major concern of researchers is the possibility of misuse or misinterpretation of their data, and this is reported as well in surveys [69,83–84]. Traditionally, research data has been shared through professional networks and by personal request [32,36,38]. These 'traditions' were incorporated into research processes as early-career researchers were indoctrinated by mentors and senior researchers [52]. This approach allowed those who owned datasets to scrutinize requests and all aspects of the requestor, including the reputation of their institution, their publications, and any other factors they felt important. Data producers had a hand in assuring their work and intellectual property were protected, privacy and confidentiality were safe, and it allowed them to exercise caution if there were any concerns around the misuse of their data, including the option to decline the request to share. Currently, funding agencies and journals are moving researchers in the direction of sharing data which is not embraced by all members of the research community [12,40,69,83–84]. In a recent paper, Campbell and colleagues [85] identified senior researchers as less likely to support data sharing while their early-career colleagues were more willing to make their data available for reuse. Researchers describe shifting to a culture of open data as a gradual transition in our findings, and stage of career may contribute to this need for a gradual shift.

Incentives were also identified as necessary for researchers within the research process that promoted open data [47,51,86]. In 2016, more than 500 researchers that received grants from the Wellcome Trust (welcome.ac.uk) in the United Kingdom were surveyed and although over half indicated that they made their data available for reuse, few reported direct benefits [69]. The lack of benefits appeared in our results and were identified as necessary yet lacking in the realm of data sharing. Suggestions for incentives included offering research grants that focused specifically on the reuse of data generated from earlier grants [51], and creating systems that ensure credit is awarded to data generators [87–90]. In one example, Pierce and colleagues [88] proposed creating enduring links between those who generate data and any time it was used in the future. This would involve linking persistent identifier (PIDs) to all datasets and provide infrastructure to link the identifiers to publications. In this strategy, data authorship would be listed on curriculum vitae, considered in academic institutions promotions criteria, and be considered by granting agencies as an element for review for funding.

There is a global movement towards openness in research that includes open data. Data sharing and reuse is a key part of this movement and anticipated benefits include promoting research transparency, verification of findings, and gaining new insights from re-analysis [8]. Despite this, it has not become a common practice [13–16]. Investing in strategies that improve skills amongst researchers that focus on improving data integrity in repositories and identifying incentives that provide motivation for data sharing are essential.

## Limitations

Quality assessment indicated that some items in the CASP tool were addressed poorly in the studies. This included author reflexivity, analysis, and ethical issues. Limitations set by journals (i.e., word counts) may restrict authors from providing rich data and thick descriptions which are characteristic of qualitative studies. Studies based on low reporting quality were not excluded as this may have eliminated those with highly relevant and insightful results [77] and were used to judge the relative contribution in developing explanations in the study findings.

The qualitative data collected for this review originates from multiple disciplines and each may use a variety of data collection methods and research processes. However, when examining the studies by discipline, over a third of the studies in our review are listed as 'combined' (37% or 17 out of 45) [91] (i.e., participants came from multiple disciplines) yet none of the authors reported this as an issue in their analysis or impacting their results (S6 Appendix). Similarly, one of the study authors (LP) conducted focus groups with academic researchers in the area of research data management (including data sharing) and found that data saturation was reached after conducting four focus groups despite collecting data from discrete disciplines (i.e., health science, humanities, natural science) [81]. While diverse tools and methods may be employed by researchers in distinct disciplines to conduct their studies, issues related to research data management were identified as a commonality [32,38,39,44–45, 47,52–54,56,58,59–60,65,70–71,75,81]

Most of the studies accepted into our review are interviews (82% or 37 out of 45 studies). While the group setting of a focus group may prompt ideas and memories from group members by listening to other participants [92], interviews provide the opportunity to go deeper into a topic and gather in-depth information [93]. When Guest and colleagues [93] performed a randomized controlled trial comparing focus groups and interviews, they found that individual interviews were more effective at generating a broad range of items at an individual level [93].

## Conclusions

Misuse and misinterpretation of data is a significant concern amongst researchers when sharing their data. Preparation of data so that it is truly reusable requires an investment in time and resources as well as skills that researchers indicate they lacked. Deficiencies in infrastructure may hamper sharing data effectively, particularly sensitive data. The availability of data is marked by researchers' decision making around what they determine is worth sharing. Currently, there is a lack of incentives for researchers to share their data with regards to academic appointment, promotion, recognition, and rewards. As such, enhancements need to be considered that focus on providing direct benefits to researchers who share their data. Identifying appropriate incentives may help improve motivation to share data and enhance the integrity of data put into repositories.

## Supporting information

**S1 Appendix. PRISMA checklist.**
(DOC)

**S2 Appendix. MEDLINE search strategy.**
(DOCX)

**S3 Appendix. Literature databases searched.**
(DOCX)

**S4 Appendix. CASP (critical appraisal skills programme): Qualitative checklist.**
(DOCX)

**S5 Appendix. Included studies.**
(DOCX)

**S6 Appendix. Included studies by discipline.**
(DOCX)

## Acknowledgments

We thank Jordan Raghunandan for collating information after the data was coded.

## Author Contributions

**Conceptualization:** Erik Blondal.

**Data curation:** Laure Perrier, Erik Blondal, Heather MacDonald.

**Formal analysis:** Laure Perrier, Erik Blondal, Heather MacDonald.

**Investigation:** Laure Perrier, Erik Blondal, Heather MacDonald.

**Methodology:** Laure Perrier, Erik Blondal.

**Project administration:** Laure Perrier.

**Writing – original draft:** Laure Perrier.

**Writing – review & editing:** Erik Blondal, Heather MacDonald.

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
