## [Decision Letter · Decision Letter 0]

4 Oct 2019

PONE-D-19-18302

The views, perspectives, and experiences of academic researchers with data sharing and reuse: a meta-synthesis

PLOS ONE

Dear Dr. Perrier,

Thank you for submitting your manuscript to PLOS ONE. After careful consideration, we feel that it has merit but does not fully meet PLOS ONE’s publication criteria as it currently stands. Therefore, we invite you to submit a revised version of the manuscript that addresses the points raised during the review process.

The reviewers found the manuscript valuable; however, several things need to be addressed before it meets PLOS publication criteria. The most immediate is the need for additional analysis, to ensure the robustness of the results. There are also some minor concerns about the legibility of figures. I would encourage the authors to address all concerns raised in the reviews. 

We would appreciate receiving your revised manuscript by Nov 18 2019 11:59PM. To enhance the reproducibility of your results, we recommend that if applicable you deposit your laboratory protocols in protocols.io, where a protocol can be assigned its own identifier (DOI) such that it can be cited independently in the future. For instructions see: http://journals.plos.org/plosone/s/submission-guidelines#loc-laboratory-protocols

We look forward to receiving your revised manuscript.

Kind regards,

Cassidy Rose Sugimoto, Ph.D.

Academic Editor

PLOS ONE

Journal Requirements:

Additional Editor Comments (if provided):

Reviewers' comments:

Reviewer's Responses to Questions

**Comments to the Author**

1. Is the manuscript technically sound, and do the data support the conclusions?

Reviewer #1: Yes

Reviewer #2: Yes

2. Has the statistical analysis been performed appropriately and rigorously? 

Reviewer #1: I Don't Know

Reviewer #2: I Don't Know

3. Have the authors made all data underlying the findings in their manuscript fully available?

Reviewer #1: Yes

Reviewer #2: Yes

4. Is the manuscript presented in an intelligible fashion and written in standard English?

Reviewer #1: Yes

Reviewer #2: Yes

5. Review Comments to the Author

Reviewer #1: 1. Is the manuscript technically sound, and do the data support the conclusions?

The use of CASP and the supporting Appendices provided the necessary information to ensure that the research was conducted in a technically sound way. The findings are described thoroughly and are supported by the data. The appendices and the quotations from the articles make it easier to determine if the data supports the conclusions.

2. Has the statistical analysis been performed appropriately and rigorously?

No statistical analysis was conducted in this study; however, the independent coders used the appropriate metric CASP to ensure that the documents were of high quality. However, surprisingly, the authors did not describe or include any intercoder reliability statistics for either the CASP coding nor the analytical themes coding. A discussion of this should be included in the study.

3. Have the authors made all data underlying the findings in their manuscript fully available?

Yes, the data is provided in a repository. Additionally, the authors provided a list of all manuscripts analyzed and the databases that were used in the appendices.

4. Is the manuscript presented in an intelligible fashion and written in standard English?

Yes, the language used in the manuscript is intelligible, with very few copyediting mistakes. There are occasionally lack of commas and other minor copyediting errors. However, the language throughout is vague, I would suggest editing throughout the make the paper more specific.

5. Review Comments to the Author

Other comments:

Figure 2 was nearly impossible to see, a higher quality image needs to be provided.

While this paper is well written and easy to follow, the findings leave the reading wanting to know why this study is important. It provides a summary of the other studies in data sharing, however, does not provide an understanding outside what is already known from the current data sharing research. I would suggest adding more to the discussion and conclusion to indicate why this study is important and what the academic community and data sharers, as well as repositories managers should learn from it.

Reviewer #2: This is an interesting “meta-synthesis” of 45 studies previously published that addressed data sharing. The conclusions are not surprising, and the results (the ‘synthesis’) has all been published previously, as well as in review articles about the same topic. This is not novel, nor are new suggestions made for further work to be done or solutions to be tested. Nevertheless, it represents a comprehensive review of 45 studies.

While the authors have aggregated concerns and opinions about sharing data, they provide no statistical analysis of the frequency of those concerns. This is a major limitation of the work. How often did a concern or opinion need to be observed in the meta-analysis in order for it to be listed as a major concern? Did any concern(s) rise above others or for different types of research data? What weight should be assigned to the importance of each of these many concerns?

While a majority of the studies utilized interviews as the principal methodology, 8 studies did not. Were there any differences between focus groups (or other) and interview results? Were any surveys found?

The authors appear to comingle studies that interviewed (or reported on focus groups of) researchers that are required to (or encouraged to) deposit data as well as attitudes of others (e.g. data scientists, students, data users, lawyers, etc.). One would anticipate that the opinions of each of these would be quite different, and different again from participants in clinical trials or research. How these different subgroups weighted their concerns, and how analyzed should be discussed. This may require incremental analysis by the authors.

It is surprising to this reviewer that there is no differentiation between the attitudes of scientists in the different fields: Investigators in some research disciplines are accustomed to sharing data (e.g. astronomy, genetics) and have standards in place for doing so. Researchers in other disciplines, most notably in human participant research where issues of privacy and confidentiality are central concerns, appear to drive reticence to share. The authors state that 17 of 45 studies were from “mixed” disciplines; nevertheless, this reviewer requests an analysis by type of research, and by ‘concern’.

The authors mention that one of the authors “conducted focus groups with academic researchers…and found that data saturation was reached after conducting four focus groups despite collecting data from discrete disciplines (i.e., health science, humanities, natural science).” How did that research factor into this current analysis and what significance does it hold?

Did the authors perceive any differences in opinions and attitudes over time?

Figure 2 is illegible. Why not present in a tabular form?

6. PLOS authors have the option to publish the peer review history of their article (what does this mean?). If published, this will include your full peer review and any attached files.

**Do you want your identity to be public for this peer review?** For information about this choice, including consent withdrawal, please see our Privacy Policy

Reviewer #1: No

Reviewer #2: No

While revising your submission, please upload your figure files to the Preflight Analysis and Conversion Engine (PACE) digital diagnostic tool, https://pacev2.apexcovantage.com/ PACE helps ensure that figures meet PLOS requirements. To use PACE, you must first register as a user. Registration is free. Then, login and navigate to the UPLOAD tab, where you will find detailed instructions on how to use the tool. If you encounter any issues or have any questions when using PACE, please email us at figures@plos.org. Please note that Supporting Information files do not need this step.

---

## [Author Response · Author response to Decision Letter 0]

11 Oct 2019

Response to Peer Reviewers

Reviewer #1

No statistical analysis was conducted in this study; however, the independent coders used the appropriate metric CASP to ensure that the documents were of high quality. However, surprisingly, the authors did not describe or include any intercoder reliability statistics for either the CASP coding nor the analytical themes coding. A discussion of this should be included in the study. 

Completed: Page 4-5

We added a new section under Materials and Methods entitled ‘Types of Studies’ and provided the references and support to outline more information on metasynthesis as a study type. This includes the clarification that metasyntheses is a qualitative review and therefore does not use quantitative approaches/analysis, e.g., statistics (Walsh & Downe 2005).

Other comments:

Figure 2 was nearly impossible to see, a higher quality image needs to be provided. 

Completed: Figure 2

The font size and dimensions have been increased on the image.

While this paper is well written and easy to follow, the findings leave the reading wanting to know why this study is important. It provides a summary of the other studies in data sharing, however, does not provide an understanding outside what is already known from the current data sharing research. I would suggest adding more to the discussion and conclusion to indicate why this study is important and what the academic community and data sharers, as well as repositories managers should learn from it. 

Completed: Page 3-4; Page 27; Page 29

Page 3-4: A paragraph was added to the Introduction to clarify why this study is important. 

Page 27, Last paragraph of Discussion: A paragraph was added to confirm why this study is important.

Page 29, Last sentence of Conclusions: The Conclusions were enhanced to clarify why this study is important. 

Reviewer #2 Comments 

This is an interesting “meta-synthesis” of 45 studies previously published that addressed data sharing. The conclusions are not surprising, and the results (the ‘synthesis’) has all been published previously, as well as in review articles about the same topic. This is not novel, nor are new suggestions made for further work to be done or solutions to be tested. Nevertheless, it represents a comprehensive review of 45 studies. 

No Changes Made

We searched for a review on this topic before undertaking our study but were not able to locate any. Reviewer #2 states “…the results (the ‘synthesis’) has all been published previously, as well as review articles about the same topic”. 

• We would be very appreciative if the citations of the qualitative reviews on the perspectives/experiences of academic researchers and data sharing could be shared to determine if there is overlap with our study

While the authors have aggregated concerns and opinions about sharing data, they provide no statistical analysis of the frequency of those concerns. This is a major limitation of the work. How often did a concern or opinion need to be observed in the meta-analysis in order for it to be listed as a major concern? Did any concern(s) rise above others or for different types of research data? What weight should be assigned to the importance of each of these many concerns? 

Completed: Page 4-5

We added a new section under Materials and Methods entitled ‘Types of Studies’ and provided the references and support to outline more information on metasynthesis as a study type. This includes the clarification that metasyntheses is a qualitative review and therefore does not use quantitative approaches/analysis, e.g., statistical analysis, meta-analysis, or weighting, i.e., quantitative approaches (Zimmer 2006; Walsh & Downe 2005; Sandelowski & Barroso 2007). 

While a majority of the studies utilized interviews as the principal methodology, 8 studies did not. Were there any differences between focus groups (or other) and interview results? Were any surveys found? 

Completed: Page 4-5

We added a new section under Materials and Methods entitled ‘Types of Studies’ and provided the references and support to outline more information on metasynthesis as a study type. This includes the clarification that metasyntheses is a qualitative review and therefore does not use quantitative approaches/analysis, e.g., subgroup analyses (Zimmer 2006; Walsh & Downe 2005; Sandelowski & Barroso 2007).

Surveys are eligible if they included mixed methods and is described on Page 5-6 under Eligibility Criteria: “Mixed methods studies that used both qualitative and quantitative methods within the same study were eligible if the qualitative portion met our inclusion criteria.”

The authors appear to comingle studies that interviewed (or reported on focus groups of) researchers that are required to (or encouraged to) deposit data as well as attitudes of others (e.g. data scientists, students, data users, lawyers, etc.). One would anticipate that the opinions of each of these would be quite different, and different again from participants in clinical trials or research. How these different subgroups weighted their concerns, and how analyzed should be discussed. This may require incremental analysis by the authors. 

Completed: Page 4-5

We added a new section under Materials and Methods entitled ‘Types of Studies’ and provided the references and support to outline more information on metasynthesis as a study type. This includes the clarification that metasyntheses is a qualitative review and therefore does not use quantitative approaches/analysis, e.g., subgroup analyses (Zimmer 2006; Walsh & Downe 2005; Sandelowski & Barroso 2007).

We stated on Page 5 that “50% or more of the total sample had to be researchers from academic institutions in order to be eligible for inclusion”. As a result, the majority of participants being analysed would be researchers (i.e., the other participants would be present in lesser numbers).

It is surprising to this reviewer that there is no differentiation between the attitudes of scientists in the different fields: Investigators in some research disciplines are accustomed to sharing data (e.g. astronomy, genetics) and have standards in place for doing so. Researchers in other disciplines, most notably in human participant research where issues of privacy and confidentiality are central concerns, appear to drive reticence to share. The authors state that 17 of 45 studies were from “mixed” disciplines; nevertheless, this reviewer requests an analysis by type of research, and by ‘concern’. 

Completed: Page 4-5

We added a new section under Materials and Methods entitled ‘Types of Studies’ and provided the references and support to outline more information on metasynthesis as a study type. This includes the clarification that metasyntheses is a qualitative review and therefore does not use quantitative approaches/analysis, e.g., sub-group analysis such as by type of research and by ‘concern’ (Walsh & Downe 2005).

The authors mention that one of the authors “conducted focus groups with academic researchers…and found that data saturation was reached after conducting four focus groups despite collecting data from discrete disciplines (i.e., health science, humanities, natural science).” How did that research factor into this current analysis and what significance does it hold? 

Completed: Page 27-28

Further explanation has been provided to this statement in order to provide clarification that despite different research methods and processes being used in various disciplines, researchers aligned with issues in research data management.

Did the authors perceive any differences in opinions and attitudes over time? 

Completed: Page 4-5

We added a new section under Materials and Methods entitled ‘Types of Studies’ and provided the references and support to outline more information on metasynthesis as a study type. This includes the clarification that metasyntheses is a qualitative review and therefore does not use quantitative approaches/analysis (Glenton 2014). 

Figure 2 is illegible. Why not present in a tabular form? Completed: Figure 2

The font size and dimensions have been increased on the image.

References

Glenton C, Lewin S. Using evidence from qualitative research to develop WHO guidelines. WHO Handbook for Guideline Development. 2nd Edition. Geneva: WHO, 2014.

Sherwood, G. (1999). Meta-synthesis: Merging qualitative studies to develop nursing knowledge. International 

Journal for Human Caring, 3, 37-42

Sandelowski M, Barroso J. Handbook for synthesizing qualitative research. New York, NY: Springer, 2007.

Walsh D, Downe S. Meta-synthesis method for qualitative research: A literature review. Journal of Advanced Nursing. 2005; 50:204-211.

Zimmer L. Qualitative meta-synthesis: A question of dialoguing with texts. Journal of Advanced Nursing. 2006; 53:311-318.

---

## [Decision Letter · Decision Letter 1]

3 Feb 2020

The views, perspectives, and experiences of academic researchers with data sharing and reuse: a meta-synthesis

PONE-D-19-18302R1

Dear Dr. Perrier,

We are pleased to inform you that your manuscript has been judged scientifically suitable for publication and will be formally accepted for publication once it complies with all outstanding technical requirements.

With kind regards,

Pablo Dorta-González, Ph.D.

Academic Editor

PLOS ONE

Additional Editor Comments (optional):

Reviewers' comments:

Reviewer's Responses to Questions

**Comments to the Author**

1. If the authors have adequately addressed your comments raised in a previous round of review and you feel that this manuscript is now acceptable for publication, you may indicate that here to bypass the “Comments to the Author” section, enter your conflict of interest statement in the “Confidential to Editor” section, and submit your "Accept" recommendation.

Reviewer #1: All comments have been addressed

Reviewer #2: (No Response)

2. Is the manuscript technically sound, and do the data support the conclusions?

Reviewer #1: Yes

Reviewer #2: Partly

3. Has the statistical analysis been performed appropriately and rigorously? 

Reviewer #1: N/A

Reviewer #2: No

4. Have the authors made all data underlying the findings in their manuscript fully available?

Reviewer #1: Yes

Reviewer #2: Yes

5. Is the manuscript presented in an intelligible fashion and written in standard English?

Reviewer #1: Yes

Reviewer #2: Yes

6. Review Comments to the Author

Reviewer #1: The authors have sufficiently addressed all of the comments from the previous review and have added additional discussion of the methods, as well as a more thorough discussion of the importance of the study.

There are some very minor types throughout and the paper should be proofread one last time before publication.

Reviewer #2: The authors have not adequately addressed the comments of either of the two reviewers, both of whom had substantive concerns with the data, the presentation of the data, and the conclusion drawn. Further, both reviewers expressed concern that there were no to limited new observations or insights from this study. Observations are thematic, and not novel. Perhaps it is more appropriate for a more specialized journal.

7. PLOS authors have the option to publish the peer review history of their article (what does this mean?). If published, this will include your full peer review and any attached files.

**Do you want your identity to be public for this peer review?** For information about this choice, including consent withdrawal, please see our Privacy Policy

Reviewer #1: No

Reviewer #2: No

---

## [Editor Report · Acceptance letter]

18 Feb 2020

PONE-D-19-18302R1 

The views, perspectives, and experiences of academic researchers with data sharing and reuse: a meta-synthesis 

Dear Dr. Perrier:

I am pleased to inform you that your manuscript has been deemed suitable for publication in PLOS ONE. Congratulations! Your manuscript is now with our production department. 

With kind regards,

on behalf of

Mr. Pablo Dorta-González 

Academic Editor

PLOS ONE